# Common bacterial infections and risk of incident cognitive decline or dementia: a systematic review protocol

Rutendo Muzambi,[1] Krishnan Bhaskaran,[1] Carol Brayne,[2] Liam Smeeth,[1] Charlotte Warren-Gash[1]

[1]Non-communicable Disease Epidemiology, London School of Hygiene and Tropical Medicine, London, UK
[2]Institute of Public Health, University of Cambridge, Cambridge, UK

**Correspondence to**
Rutendo Muzambi;
rutendo.muzambi@lshtm.ac.uk

## ABSTRACT

**Introduction** The global burden of dementia is rising, emphasising the urgent need to develop effective approaches to risk reduction. Recent evidence suggests that common bacterial infections may increase the risk of dementia, however the magnitude and timing of the association as well as the patient groups affected remains unclear. We will review existing evidence of the association between common bacterial infections and incident cognitive decline or dementia.

**Methods and analysis** We will conduct a comprehensive search of published and grey literature from inception to 18 March 2019. The following electronic databases will be searched; MEDLINE, EMBASE, Global health, PsycINFO, Web of Science, Scopus, Cochrane Library, the Cumulative Index to Nursing and Allied Health Literature, Open Grey and the British Library of Electronic Theses databases. There will be no restrictions on the date, language or geographical location of the studies. We will include longitudinal studies with a common clinically symptomatic bacterial infection as an exposure and incident cognitive decline or dementia as an outcome. Study selection, data extraction and risk of bias will be performed independently by two researchers. We will assess the risk of bias using the Cochrane collaboration approach. The overall quality of the studies will be assessed using the Grading of Recommendations, Assessment, Development and Evaluations criteria. We will explore the heterogeneity of relevant studies and, if feasible, a meta-analysis will be performed, otherwise we will present a narrative synthesis. We will group the results by exposure and outcome definitions and differences will be described by subgroups and outcomes.

**Ethics and dissemination** Ethical approval will not be required as this is a systematic review of existing research in the public domain. Results will be disseminated in a peer-reviewed journal and presented at national and international meetings and conferences.

**PROSPERO registration number** CRD42018119294.

## Strengths and limitations of this study

► To the best of our knowledge, this will be the first systematic review assessing the association between common clinically symptomatic bacterial infections and the risk of incident cognitive decline and dementia in longitudinal studies.

► We will perform a comprehensive search of published and grey literature with no restrictions on date, language or geographical location.

► The review will develop existing evidence to generate better knowledge on the magnitude and direction of the relationship between common bacterial infections and subsequent cognitive decline or dementia.

► Heterogeneity in the way infections, cognitive decline and dementia are defined is expected, which could affect the feasibility of performing a meta-analysis and interpretation of findings.

► There may be difficulty ascertaining whether lower respiratory tract infections are bacterial or viral.

## INTRODUCTION
### Rationale

Dementia is a clinical syndrome that significantly contributes to disability and dependence worldwide, with a devastating impact on individuals, caregivers and healthcare services.[1–3] It is characterised by a progressive deterioration in cognition and behaviour that interferes with an individual's ability to perform activities of daily living.[4] In 2018, approximately 50 million people worldwide were estimated to be living with dementia, and this figure is projected to rise to 152 million by 2050.[5]

Age is the single biggest risk factor for dementia, with the risk doubling every 5 years after the age of 65.[6] Despite this, there have been some indications that the risk for dementia can be reduced, with more recent cohorts in the UK demonstrating a significantly lower risk at any given age.[7] It has become clear from clinical and neuropathological studies that the risk for dementia is complex, not driven by genetics, vascular factors or age alone. Given the rapidly increasing ageing population and the absence of pharmacological treatments that can delay the onset or progression of dementia, identification of effective strategies to reduce risk, as age increases, has become a public health

priority.[8] As a result, recent literature has focused on modifiable risk factors. It has been estimated that around a third of dementia cases worldwide can be attributed to modifiable risk factors.[9–11]

Bacterial infections have been identified as potential risk factors for dementia. For the past few decades, a large body of research has been published on the association between bacterial pathogens and Alzheimer's disease, particularly in postmortem brain tissue.[12–14] Despite this, the temporality of this relationship remains unclear due to the cross-sectional nature of the data collected in these studies. A recent meta-analysis of 27 serological, cerebrospinal fluid and postmortem brain studies found that infections due to *Chlamydia pneumoniae* and Spirochaetes were associated with a 5-fold (OR 5.66; 95% CI 1.83 to 17.51) and 10-fold (OR 10.61; 95% CI 3.38 to 33.29) increased risk of Alzheimer's disease, respectively.[15] However, temporality could not be assessed in these studies. The meta-analysis was further limited by the inclusion of studies with small sample sizes and the focus on Alzheimer's disease rather than all types of dementia.

Common bacterial infections are well recognised to be associated with acute changes in cognition, manifested as delirium, among older adults.[16] In turn, delirium is strongly associated with an increased risk of subsequent cognitive decline and dementia.[17–19] This raises the challenge of disentangling the relative contribution of the (potentially reversible) ageing immune system's response to acute infections, from ongoing neuropathological processes, both of which may affect cognition.

Longitudinal studies with a follow-up time sufficient enough for delirium to resolve, are important in distinguishing between delirium and long-term cognitive impairment. Additionally, longitudinal studies that compare incidence of cognitive decline or dementia in individuals with and without common bacterial infections provide evidence for the temporality and magnitude of this association. Although the prevalence of common bacterial infections, such as pneumonia and urinary tract infections, in individuals with dementia is well known,[20 21] the incidence of cognitive decline and dementia is less established.

The mechanisms by which bacterial infections may increase the risk of cognitive decline and dementia are unclear but may involve inflammation.[22 23] Bacterial infections may trigger an inflammatory response in the brain resulting in the release of pro-inflammatory mediators and activation of cytotoxic microglia. This may result in deterioration of cognitive function, possibly increasing the risk of dementia.[12]

To date, no systematic review has been performed on the incidence of cognitive decline and dementia in individuals with a common bacterial infection causing clinical illness. As these infections frequently occur in the community, early recognition, treatment or prevention of common bacterial infections could have important public health implications in reducing the burden of dementia.

## Objectives

The primary objective of the proposed systematic review is to summarise the current literature investigating the association between common clinically symptomatic bacterial infections (sepsis, lower respiratory tract infections, urinary tract infections and skin and soft tissue infections) and incident cognitive decline and dementia in longitudinal studies. A secondary objective will be to identify gaps in literature and recommendations for future research on this topic.

## METHODS

The present systematic review protocol was developed according to the Preferred Reporting Items for Systematic Reviews and Meta-analyses Protocols (PRISMA-P) statement and was registered on the PROSPERO database.[24 25] Any amendments to the protocol will be updated in PROSPERO.

We will report the systematic review in line with the PRISMA statement.[26] If a meta-analysis is feasible, it will be reported according to the Meta-analysis of Observational Studies in Epidemiology statement.[27]

### Eligibility criteria

Studies will be eligible for inclusion into the present study if they meet the inclusion criteria mentioned below.

### Study design

We will include longitudinal studies; retrospective and prospective cohort studies, case-control studies and randomised controlled trials (RCTs). We will include studies specifically investigating the association between common bacterial infections and cognitive decline or dementia. Although it would not be possible to perform RCTs specifically addressing our research question, we will consider studies derived from RCTs which could include cohort or case-control studies from an RCT data source. We will include studies in which the exposure is ascertained prior to the occurrence of the outcome events in order to investigate the temporal relationship between common bacterial infections and subsequent cognitive decline or dementia. Additionally, to avoid including studies focusing on short-term reversible changes in cognition, rather than long-term cognitive decline, we will include studies in which cognitive decline was measured at least 3 months following infection.

### Study population

We will include human studies of adults aged 18 years and over with no restrictions on the sex, ethnicity, prior health status or residence of participants. We will include studies conducted in any healthcare setting.

### Exposure

We will include studies in which exposure is defined as symptomatic illness due to common bacterial infections, either suspected clinically or confirmed by isolation of a bacterial pathogen. Types of bacterial infections will be

subdivided into sepsis, lower respiratory tract infections, skin and soft tissue infections and urinary tract infections. The possible pathophysiological mechanisms of bacterial infections on cognitive decline or dementia may differ depending on the site of infection, and thus we will only include studies assessing the independent effect of each type of infection on our outcomes. In addition, infections are likely to differ in terms of severity, particularly sepsis, which further highlights the need to assess their effect separately. We will exclude studies that focus on a specific bacterial pathogen as the exposure rather than the symptomatic disease (ie, isolation of a bacterium by PCR alone in the absence of clinical symptoms).

### Comparators

Studies eligible for inclusion will include a comparison group. For cohort studies and secondary analyses of longitudinal RCT data, individuals exposed to common bacterial infections will be compared with those unexposed to infections. For case-control studies, cases with dementia or cognitive decline will be compared with a control group without dementia or cognitive decline.

### Outcomes

We will have two primary outcomes. These will be (1) incident cognitive decline and (2) incident dementia (all types). Cognitive decline will be defined clinically. Dementia will also be defined clinically, with or without neuroimaging or histopathology. If a sufficient number of eligible studies are available, dementia will be subdivided into dementia types.

### Literature searches

We will systematically search electronic databases of published and grey literature from inception to 18 March 2019. The following databases will be searched: MEDLINE (Ovid interface), EMBASE (Ovid interface), Global health (Ovid interface), PsycINFO (Ovid Interface), Web of Science, Scopus, Cochrane Library, the Cumulative Index to Nursing and Allied Health Literature, Open Grey and the British Library of Electronic Theses databases. Additionally, we will search the reference lists of the included studies to identify any relevant articles not captured in the search strategy. We will search the databases using subject headings, where possible, and keywords related to the exposure, outcome and study design. These search terms will be combined using Boolean logical operators.

We carried out a preliminary search on PubMed to ensure we would capture a sufficient number of studies for inclusion into the review. We developed a search strategy for the MEDLINE database which is provided in online supplementary appendix 1. The search strategy was developed in consultation with a librarian at the London School of Hygiene and Tropical Medicine and was subsequently peer reviewed based on the Peer-Review for Electronic Search Strategies.[28] We will translate our search strategy in all databases using search syntaxes specific to each database. No restrictions will be placed on the geographical location, language or date of publication of the studies. Any potentially relevant non-English studies will be translated.

### Study records
#### Data management

We will import the search results into the reference manager software EndNote (X8.0.2). Duplicate entries will be identified and removed.

#### Study selection

Two researchers will independently screen all titles and abstracts against the eligibility criteria. The researchers will then independently screen the full-text articles of potentially eligible articles and decide on whether the inclusion criteria have been met. Any disagreements between the reviewers will be discussed and if necessary a third reviewer will be consulted. Reasons for exclusion of studies will be recorded. We will document the study selection process using the PRISMA flow diagram.[26] If multiple papers arise from the same population, we will include the paper with the largest sample size and most detailed information about the exposure and outcome. We pilot tested our study selection process to ensure that the inclusion criteria can be reliably applied.

#### Data collection process

Two researchers will independently extract data from included papers onto a predesigned form. If necessary, the authors will be contacted directly to obtain any missing information. We will perform pilot testing on the data extraction form to identify any missing or irrelevant criteria, and we will modify the form accordingly.

### Data items

We will use the Population, Exposure, Comparator, Outcomes and Study characteristics framework to design our data extraction form. The following information will be extracted:
1. Population: age (mean, median or range), sex, inclusion and exclusion criteria.
2. Exposure: definition of exposure, type of bacterial infection, cause of sepsis, number of exposed.
3. Comparators: identification and definition of comparator, number of comparators.
4. Outcomes: definition of outcome and identification of cognitive decline and dementia, number of participants with the outcome.
5. Study characteristics: authors, name of study, year of publication, study design, type of longitudinal study, healthcare setting, country, sample size, duration of follow-up.

Regarding the study results, we will extract unadjusted and adjusted estimates and their corresponding 95% CI for each exposure and outcome. Data on subgroup or sensitivity analyses will be extracted. We will extract additional data on antibiotic treatment, if indicated, given that antibiotics have been associated with delirium,[29]

which is in turn a risk factor for cognitive decline and dementia. We will also extract data on confounding variables. Factors considered to be potential confounders include age,[30] sex,[31] socioeconomic status,[32] ethnicity,[33] smoking,[34] alcohol consumption,[35] cardiovascular disease,[36] diabetes,[37] renal dysfunction,[38] psychiatric disorders,[39] cerebrovascular disease,[40] chronic obstructive pulmonary disease[41] and immunodeficiency disorders.[42] We will assess whether studies have adequately assessed for potential confounders as part of our risk of bias and study quality assessments.

### Risk of bias in individual studies

We will use a sample of studies to pilot test the risk of bias form to ensure the criteria can be applied consistently by both reviewers. Two researchers will independently assess the risk of bias in line with the Cochrane collaboration approach.[43 44] We will examine the risk of bias for RCTs using the following domains: sequence generation; allocation concealment; blinding of participants and personnel; blinding of outcome assessment; incomplete outcome data; selective outcome reporting and other potential threats to validity.[44] We will assess the risk of bias for observational studies using the following domains: confounding, selection of participants, misclassification of variables, missing data, reverse causation, generalisability and study power.

### Data synthesis and meta-bias(es)

We will group studies according to the outcome (cognitive decline or dementia) and exposure (type of common bacterial infection) and synthesise them narratively. Data will be summarised in predefined tables. We will consider subgroup analyses according to age group, gender, dementia subtype and risk of bias. If possible, we will explore the effect of antibiotic treatment on cognitive decline or dementia.

A meta-analysis will be considered feasible if there are a minimum of at least two studies that are homogeneous in terms of study design, type of common bacterial infection and type of outcome. We will pool effect measures (ORs, risk ratios or HRs) from the studies in order to perform the meta-analysis.

We will assess statistical heterogeneity through the use of forest plot, $\chi^2$ test and $I^2$ statistic. Depending on the level of heterogeneity, a fixed or random effects model will be selected to calculate the pooled incidence and corresponding 95% CI. A $\chi^2$ test with a p value of 0.1 will be considered statistically significant. We will consider an $I^2$ value of >25% to indicate moderate heterogeneity, which will guide the use of a random effects model.[45 46] We will assess publication bias and small study effects using funnel plots, provided that there are 10 or more studies eligible for inclusion into the meta-analysis.[47]

### Confidence in cumulative evidence

We will assess the overall quality of evidence for each outcome using the Grading of Recommendations, Assessment, Development and Evaluation tool.[48] The domains assessed will include study limitations, inconsistency, indirectness, imprecision and publication bias.[49] The strength of the evidence will be categorised as high, moderate, low and very low.

### Patient and public involvement

No patients or public were directly involved in the design of this study. However, we sought advice on the dissemination of our findings from lay volunteers assigned to Rutendo Muzambi's PhD studentship by the Alzheimer's Society.

## ETHICS AND DISSEMINATION

This systematic review will provide evidence for the role of common bacterial infections in the development of cognitive decline and dementia. The systematic review will be submitted for publication in a peer-reviewed journal and the results may be presented at national and international conferences and meetings relevant to the field. This review will highlight gaps in current literature and identify future research directions.

**Contributors** CW-G, KB and LS conceived and designed the study, revised the protocol and search strategy critically. RM conceived and designed the study, drafted and revised the protocol and developed and revised the search strategy. CB contributed to the design of the study and revised the protocol and search strategy critically. The final version of the protocol was read and approved by all authors.

**Funding** RM is supported by an Alzheimer's Society PhD studentship 379 (AS-PhD-17-013). CWG is supported by a Wellcome Intermediate Clinical Fellowship, Wellcome Trust (201440_Z_16_Z).

**Competing interests** None declared.

**Patient consent for publication** Not required.

**Ethics approval** Ethical approval will not be required as the review will summarise data from previous studies.

**Provenance and peer review** Not commissioned; externally peer reviewed.

**Data availability statement** There are no data in this work.

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
