## [Reviewer comments · BMJ Open]

ARTICLE DETAILS

TITLE (PROVISIONAL)	Common bacterial infections and risk of incident cognitive decline or dementia: a systematic review protocol
AUTHORS	Muzambi, Rutendo; Bhaskaran, Krishnan; Brayne, Carol; Smeeth, Liam; Warren-Gash, Charlotte

VERSION 1 – REVIEW

REVIEWER	Robert B. Wallace, MD, MSc College of Public Health University of Iowa USA
REVIEW RETURNED	15-Apr-2019

GENERAL COMMENTS	This is a potentially useful systematic review, and the proposed methodology is in keeping with modern standards and guidelines. There are some issues the authors may wish to consider: 1. There is almost always value in conducting a scoping exercise in advance of the review, to determine whether there is likely to be sufficient relevant literature to review. If that was performed, it should be reported in the proposal document.2. It would seem important to pay careful attention to the presence of confounders in the studies chosen for review. Perhaps most important would be those factors that might cause both infections and dementia. Comorbid conditions such as diabetes, alcoholism or immune deficiency disorders might be examples.3. In keeping with #2, some attention to the hypothetical mechanisms for an association (if any exists) would be useful. For example, direct brain infections may be different from local tissue infections.4. It is noted that all institutional settings would be studied. This should be handled carefully because chronic institutionalization might be a very different situation from community dwellers. For example, in the US, over half of nursing home residents already have dementia, and infection risk may be different because of nosocomial exposures and other clinical reasons.
---

REVIEWER	Birgit Teufer University for Continuing Education Krems (Danube University Krems), Department for Evidence-based Medicine and Clinical Epidemiology
REVIEW RETURNED	07-May-2019

GENERAL COMMENTS	First of all, thanks for the possibility to review this submitted manuscript which addresses, in my opinion, a relevant topic in dementia research.
---

	Dear authors, you provide a well written introduction for a topic in the field of dementia research that is important in my opinion. I have a few questions and comments on the planned methods. Page 5 "Study design": How could randomised controlled trials be performed on the topic of interest? Page 6 "Comparators": As you plan to include retrospective and prospective cohort studies, randomised controlled trials (?) and case control studies there will be studies with and without control groups and you should define precisely what kind of comparators you will accept. I cannot really imagine what "person time unexposed to common bacterial infections" mean; if you mean the lifetime of later exposed people before exposure there could be a methodological problem as age (= time that passes by) is one of the main risk factors. Please clarify and define precisely what you mean. Page 6 "Outcomes": what is "a sufficient number of eligible studies"? Page 6 "Literature searches": According to the PRESS-Statement a peer review of the search strategy is recommended. If the search has not been performed yet, please consider a peer review of the search strategy. (McGowan, J., Sampson, M., Salzwedel, D. M., Cogo, E., Foerster, V., & Lefebvre, C. (2016). PRESS peer review of electronic search strategies: 2015 guideline statement. Journal of clinical epidemiology, 75, 40-46.) Page 6 and 7, "Study selection process" and "data collection process": Will there be a pilot testing of review and data extraction forms? If not, why not? Page 8 "Confidence in cumulative evidence": There are four categories from high to very low in GRADE, you only mentioned high, moderate and low. As observational studies start at "low" you surely need the last category "very low" as well. Why wouldn't you include this category? I'm looking forward to reviewing a revision of this manuscript. Best regards Birgit Teufer
--	---

VERSION 1 – AUTHOR RESPONSE

Response to reviewer 1

1. There is almost always value in conducting a scoping exercise in advance of the review, to determine whether there is likely to be sufficient relevant literature to review. If that was performed, it should be reported in the proposal document.

Thank you for your comment. We carried out preliminary searches prior to conducting our systematic review to ensure there would be sufficient literature eligible for inclusion. We have now included this in the methods section on page 7.

New text:

“We carried out a preliminary search on PubMed to ensure we would capture a sufficient number of studies for inclusion into the review.”

2. It would seem important to pay careful attention to the presence of confounders in the studies chosen for review. Perhaps most important would be those factors that might cause both infections and dementia. Comorbid conditions such as diabetes, alcoholism or immune deficiency disorders might be examples.

We agree that it will be important to assess the presence of confounding. To address this, we will extract information on the confounders adjusted for in each study during the data extraction process. Additionally, as part of our risk of bias and study quality assessment, we will assess adjustment for confounding factors. We have now added new text to the methods section on page 8 to clarify this, and we have also stated examples of the potential confounding factors.

New text:

“Factors considered to be potential confounders include age,[29] sex,[30] socioeconomic status,[31] ethnicity,[32] smoking,[33] alcohol consumption,[34] cardiovascular disease,[35] diabetes,[36] renal dysfunction,[37] psychiatric disorders,[38] cerebrovascular disease,[39] chronic obstructive pulmonary disease,[40] and immunodeficiency disorders. [41] We will assess whether studies have adequately assessed for potential confounders as part of our risk of bias and study quality assessments.”

3. In keeping with #2, some attention to the hypothetical mechanisms for an association (if any exists) would be useful. For example, direct brain infections may be different from local tissue infections.

We agree that it would be useful to pay attention to the hypothetical mechanisms underlying an association between infections and dementia. The possible pathophysiological mechanisms of bacterial infections on dementia risk may differ depending on the site of infection, and thus we will only include studies assessing the independent effect of each type of infection on our outcomes. We have now added new text to the introduction and methods on pages 5 and 6.

New text:

“The mechanisms by which bacterial infections may increase the risk of cognitive decline and dementia are unclear but may involve inflammation. [22, 23] Bacterial infections may trigger an inflammatory response in the brain resulting in the release of pro-inflammatory mediators and activation of cytotoxic microglia. This may result in deterioration of cognitive function, possibly increasing the risk of dementia. [12]”

“The possible pathophysiological mechanisms of bacterial infections on cognitive decline or dementia may differ depending on the site of infection, and thus we will only include studies assessing the independent effect of each type of infection on our outcomes.”

4. It is noted that all institutional settings would be studied. This should be handled carefully because chronic institutionalization might be a very different situation from community dwellers. For example, in the US, over half of nursing home residents already have dementia, and infection risk may be different because of nosocomial exposures and other clinical reasons.

We agree that the setting in which the studies are conducted may have an effect on the risk of infections and dementia, and it will be important to address this. We acknowledge that individuals recruited from different healthcare settings may exhibit different characteristics, such as frailty or comorbidities, which may affect their risk of infections and dementia. In addition, the type and severity of infections, and the likelihood of a dementia diagnosis may also differ across a range of settings. We understand that including studies from a variety of healthcare settings will likely increase the heterogeneity of our studies, however, it may allow us to understand the impact various healthcare settings may have on the risk of dementia and may provide insights into the population groups most at risk of dementia. To address this, we will capture information on the study setting during the data extraction process and we will discuss the possible impact of this in the review.

Response to reviewer 2

1. How could randomised controlled trials be performed on the topic of interest?

Although it would not be possible to perform randomised controlled trials (RCTs) specifically on our research question, we will include studies which used data derived from RCTs. This could include cohort or case control studies from an RCT data source. We have now added new text on page 5 and 6 to clarify this.

New text:

"We will include longitudinal studies; retrospective and prospective cohort studies, case control studies and randomised controlled trials (RCTs). We will include studies specifically investigating the association between common bacterial infections and cognitive decline or dementia will be eligible for inclusion. Although it would not be possible to perform randomised controlled trials specifically addressing our research question, we will consider studies derived from RCTs which could include cohort or case control studies from an RCT data source."

2. "Comparators": As you plan to include retrospective and prospective cohort studies, randomised controlled trials (?) and case control studies there will be studies with and without control groups and you should define precisely what kind of comparators you will accept. I cannot really imagine what "person time unexposed to common bacterial infections" mean; if you mean the lifetime of later exposed people before exposure there could be a methodological problem as age (= time that passes by) is one of the main risk factors. Please clarify and define precisely what you mean.

Thank you for your suggestion to clarify how comparators will be defined in each study. We have now updated our methods section on page 6 to clarify how each comparator will be defined in each study design.

New text:

"Studies eligible for inclusion will include a comparison group. For cohort studies and secondary analyses of longitudinal RCT data, individuals exposed to common bacterial infections will be compared to those unexposed to infections. For case control studies, cases with dementia or cognitive decline will be compared to a control group without dementia or cognitive decline."

3. Page 6 "Outcomes": what is "a sufficient number of eligible studies"?

If we have minimum of two studies that are homogenous in terms of exposure, outcome and study design we will perform a meta-analysis. We have now clarified this in our methods section on page 8.

New text:

“A meta-analysis will be considered feasible if there are a minimum of at least two studies that are homogeneous in terms of study design, type of common bacterial infection and type of outcome.”

4. Page 6 "Literature searches": According to the PRESS-Statement a peer review of the search strategy is recommended. If the search has not been performed yet, please consider a peer review of the search strategy. (McGowan, J., Sampson, M., Salzwedel, D. M., Cogo, E., Foerster, V., & Lefebvre, C. (2016). PRESS peer review of electronic search strategies: 2015 guideline statement. *Journal of clinical epidemiology*, 75, 40-46.)

The search strategy was developed in consultation with a librarian at the London School of Hygiene and Tropical Medicine and was subsequently peer reviewed. We have now stated this in our methods section on page 7.

New text:

“We developed a search strategy for the MEDLINE database which is provided in online supplementary appendix 1. The search strategy was developed in consultation with a librarian at the London School of Hygiene and Tropical Medicine and was subsequently peer reviewed based on the Peer-Review for Electronic Search Strategies (PRESS). [28]”

5. Page 6 and 7, "Study selection process" and "data collection process": Will there be a pilot testing of review and data extraction forms? If not, why not?

We pilot tested our study selection process to ensure that our inclusion criteria could be reliably applied and that studies could be classified appropriately. We will perform pilot testing on the data extraction and risk of bias forms to identify any data that are missing or likely to be irrelevant. We have now included this in our methods section on pages 7 and 8.

New text page 7:

“We pilot tested our study selection process to ensure that the inclusion criteria can be reliably applied.”

New text page 7:

“We will perform pilot testing on the data extraction form to identify any missing or irrelevant criteria, and we will modify the form accordingly.”

New text page 8:

“We will use a sample of studies to pilot test the risk of bias form to ensure the criteria can be applied consistently by both reviewers.”

6. "Confidence in cumulative evidence": There are four categories from high to very low in GRADE, you only mentioned high, moderate and low. As observational studies start at "low" you surely need the last category "very low" as well. Why wouldn't you include this category?

Thank you for your suggestion. There are indeed four categories in GRADE and we have now included the 'very low' category to our methods section on page 9.

New text:

“The strength of the evidence will be categorised as high, moderate, low and very low.

VERSION 2 – REVIEW

REVIEWER	Robert B. Wallace, MD, MSc Departments of Epidemiology and Internal Medicine University of Iowa, USA
REVIEW RETURNED	16-Jun-2019

GENERAL COMMENTS	The authors have been responsive to comments, and assuming there are a suitable number of studies, this should succeed. There are few suggestions, just for the authors' consideration: 1. Expanding the infectious causes a bit may be helpful, such as joint infection or severe periodontitis. 2. Some antibiotics have been associated with delirium and possibly dementia. There might be value in exploring the studies as to whether infections treatments were noted. 3. The issue of nosocomial infections is probably worth exploring, if they are indicated. These may be an indicator of prolonged institutionalization, or more chronic infections. 4. Some studies may have included sepsis, which has been shown to be associated with cognitive change. Sepsis may require separate consideration because of its severity.
--

REVIEWER	Birgit Teufer University for Continuing Education Krems (Danube University Krems), Department for Evidence-based Medicine and Clinical Epidemiology, Austria
REVIEW RETURNED	25-Jun-2019

GENERAL COMMENTS	All my comments have been taken into account and the manuscript has been amended accordingly.
---

VERSION 2 – AUTHOR RESPONSE

Response to reviewer 1

1. Expanding the infectious causes a bit may be helpful, such as joint infection or severe periodontitis.

Thank you for your comment. Although other infectious causes may be associated with cognitive decline or dementia, we aim to specifically focus on common bacterial infections. We recognise that joint infections and severe periodontitis may lead to sepsis and as such we will extract data on the causes of sepsis, if indicated. We have now included this to the methods section on page 7.

New text:

“Exposure: definition of exposure, type of bacterial infection, cause of sepsis, number of exposed.”

2. Some antibiotics have been associated with delirium and possibly dementia. There might be value in exploring the studies as to whether infections treatments were noted.

We agree that certain antibiotics have been associated with delirium. Our outcome of interest is long-term cognitive decline, rather than short-term transient manifestations.

As a result, we will only include studies in which cognitive decline was ascertained at least 3 months following infection, to avoid including studies focusing on delirium. However, we will capture information on antibiotic treatment and consider exploring the effect of antibiotics on cognitive decline or dementia, if possible. We have now reworded and revised our methods section on page 6 and 8.

New text:

“Additionally, to avoid including studies focusing on short term reversible changes in cognition, rather than long term cognitive decline, we will include studies in which cognitive decline was measured at least 3 months following infection.”

“We will extract additional data on antibiotic treatment, if indicated, given that antibiotics have been associated with delirium,[29] which is in turn a risk factor for cognitive decline and dementia.”

“If possible, we will explore the effect of antibiotic treatment on cognitive decline or dementia.”

3. The issue of nosocomial infections is probably worth exploring, if they are indicated. These may be an indicator of prolonged institutionalization, or more chronic infections.

We appreciate your comment on exploring nosocomial infections. Nosocomial infections may be a marker of prolonged hospitalisation as well as frailty and comorbidities. These infections may also be associated with more chronic or severe infections. However, we feel that to explore this fully would be beyond the scope of this review. Although we will extract information on healthcare setting, it is likely that the majority of people hospitalised with infections will have acquired the infection in the community. It is unlikely that these studies will present information on whether these infections were acquired in hospital or in the community. Therefore, addressing this fully would possibly require a new original research study, specifically looking at this question.

4. Some studies may have included sepsis, which has been shown to be associated with cognitive change. Sepsis may require separate consideration because of its severity.

We agree that sepsis has been associated with cognitive impairment and we will address the effect of each type of infection (sepsis, lower respiratory tract infections, skin and soft tissue infections and urinary tract infections) separately. We have now added new text to the methods section on page 6 to clarify this further.

New text:

“In addition, infections are likely to differ in terms of severity, particularly sepsis, which further highlights the need to assess their effect separately.”